# Waste-to-Energy Recovery from Municipal Solid Waste: Global Scenario and Prospects of Mass Burning Technology in Brazil

**Natália Dadario** [1] , **Luís Roberto Almeida Gabriel Filho** [1] , **Camila Pires Cremasco** [1] , **Felipe André dos Santos** [1] , **Maria Cristina Rizk** [2] **and Mario Mollo Neto** [1,*]

[1] School of Sciences and Engineering, São Paulo State University (UNESP), Tupã 17602-496, Brazil
[2] School of Sciences and Technology (FCT), São Paulo State University—UNESP, Presidente Prudente 19060-900, Brazil
* Correspondence: mario.mollo@unesp.br; Tel.: +55-14-3404-4200

**Abstract:** Inadequate disposal of Municipal Solid Waste (MSW) is one of the greatest environmental issues confronted nowadays. One of the techniques used for its final disposal is incineration, otherwise known as mass burning. Although this procedure remains very controversial in Brazil, some recent studies published in Europe reveal a large amount of misinformation about it. It has been widely used in European countries, Japan, and a few U.S. cities and has been increasingly and significantly adopted in China. Therefore, this article aims to carry out a literature review on the evolution of waste-to-energy recovery from Municipal Solid Waste (MSW) worldwide and the progress of mass-burning technologies, particularly in the Brazilian context. For such a purpose, global scientific databases were selected and some of their results allowed us to present how the main WtE recovery technologies function, as well as their benefits and impacts. Moreover, it was possible to systematize the main regulatory frameworks on the theme in Brazil and to reveal the country's electricity generation capacity, in addition to depicting the progress of Waste-to-Energy Plants (WtEPs) undergoing licensing processes in the state of São Paulo.

**Keywords:** waste heat recovery; energy recovery; waste-to-energy; mass burning; municipal solid waste

## 1. Introduction

Incineration, also known as mass burning, is a well-established method offering the main advantage of energetically recovering non-biodegradable and low-moisture materials worldwide [1], as there is no need for any treatment process or prior processing [2].

In addition to incineration, there are two other technologies for heat recovery from municipal solid waste (MSW), i.e., pyrolysis and gasification. However, these technologies are still under investigation and there is no large-scale use for them; therefore, they are unfeasible for commercial purposes [3]. Thus, incineration is the most reliable and cost-effective technology for waste-to-energy recovery from municipal solid waste (MSW) on the market; moreover, it is the most widely used globally because there are over 2200 plants in operation in more than 45 nations [4].

One of its main disadvantages refers to the environmental risks entailed by the emission of hazardous pollutants when not performed properly, which can in turn cause damage to human health and the environment itself [5,6]. However, when performed following legislation establishing emission limits, the process becomes safe and even recommended by the Intergovernmental Panel on Climate Change (IPCC), as it uses a smaller implementation area than a sanitary landfill does, in addition to being quite advantageous to large urban centers since it also leads to lower transportation costs. Furthermore, it enables more efficient waste-to-energy recovery, as it replaces fossil energy sources and optimizes the use of natural resources; ultimately, it also reduces greenhouse gas (GHG) emissions [7].

Data from the 5th IPCC Report reveal that landfill gas plants can capture only 50% of methane at best. In the case of inadequate waste disposal, such as dumps, GHG

emissions are even higher in controlled landfills and sanitary landfills that fail to capture and incinerate biogas [8].

Thus, despite the capital investment required to implement a sanitary landfill able to capture 30% less methane than a Waste-to-Energy (WtE) plant, WtE plants are generally more cost-effective along their lifespan of 30 years or so, due to increased electricity production [9].

In the current Brazilian scenario, incineration is the most commonly studied technique among WtE technologies, mainly because it is already very traditional and well established in several countries worldwide, in addition to presenting optimal benefit–cost ratio, as aforementioned [8,10,11].

In Brazil, there are no WtE plants in operation on a commercial scale fed by MSW to date [12], but there are Waste-to-Energy Plant (WtEP) projects in the municipalities of Barueri and Mauá, both in the state of São Paulo. As for the one in Barueri, 825 tons of MSW are expected to be processed daily with 17 megawatts (MW) of installed power, while that in Mauá has 77 MW and processes 3000 tons/day of waste [4].

Given the above, this research aims to carry out a literature review on the evolution of WtE recovery from Municipal Solid Waste (MSW) in the world and explore the advancement of mass-burning technologies, particularly in the Brazilian context. The article is divided into four sections. Section 2 presents the scenario in which WtE technologies are being developed worldwide and their current stage of development in several countries. Section 3 focuses on Brazilian energy use, particularly the mass-burning technology which is currently the most cost-effective technique. In addition, it presents regulatory frameworks regarding the theme, the current Brazilian power generation capacity, and Waste-to-Energy Plants (WtEPs) undergoing licensing processes in the state of São Paulo. Finally, the last section draws final considerations.

## 2. Literature Review

### 2.1. Different Technologies Used in Waste-to-Energy Recovery

All methods of heat treatment of waste with energy recovery, as well as waste fuels, are collectively referred to as Waste-to-Energy (WtE) [13]. WtE technologies include biological and thermochemical conversion systems [14].

As the main thermochemical conversion systems, there are the processes of (a) incineration, also called mass burning, (b) pyrolysis, and (c) gasification, which differ mainly due to the amount of oxygen present in the reaction medium. Mass-burning incineration operates with excess oxygen, while in gasification combustion occurs partially, that is, with oxygen deficit, and pyrolysis with a total absence of it [15].

The most used form of the combustion process is complete oxidation (mass burning), that is, burning USRs in designed ovens [13]. Data from the Intergovernmental Panel on Climate Change [8] confirm the predominance of the technique by pointing out that 90% of the WtE plants in the world are of the type combustion mass burning with mobile grid because this is the most cost-effective method today.

Other treatment technologies, such as gasification and pyrolysis, are still very uncommon worldwide because they are complex technologies. The first, for example, requires a drying pretreatment of the US [7] and the second needs an external power source [13]. These additional costs diminish its competitiveness in the face of mass-burning technology. However, despite the financial unfeasibility of these techniques for many contexts, many advances have been made in these new heat treatment technologies, especially in Japan, which has been the world leader in the development and application of these non-traditional treatments, with more than 100 plants for these relatively new processes [4].

Mass-burning technology, also called incineration, can be defined as a thermochemical process that through the oxidation of USRs, in which furnaces are subjected to high temperatures, between 750 °C and 1100 °C, and with the presence of oxygen under stoichiometric or excess conditions, aims to decrease the volume and mass of waste, extending the life of landfills [11,15–18]. It is currently possible to reduce the initial volume of USRs by

90% and their mass by 75%, depending on the composition and degree of recovery of the materials [19].

The great advantage of the technique, given other thermal processes, is that it can accept a wide variety of waste, of various sizes and sources [20]. In addition to the destination of the US, the process also has the benefit of power generation.

In this process, oxygen reacts with combustible elements present in the waste, such as carbon, oxygen, and sulfur, converting chemical energy into heat [21]. In addition to heat production, it is possible to produce electricity through the heating energy of the materials [22]. In this process, the generation of energy occurs after the generation of steam in boilers, which is sent to the turbines resulting in electricity [17,18].

The generation of electricity by the incineration of RSU is similar to the process of conventional thermal power plants of Cycle Rankine, in which the vapors generated in the boilers are driven using steam turbines that drive electric generators that produce electricity [2,18]. The generation capacity will depend on the efficiency of the process of transformation of heat into electric energy and the calorific value of the incinerated material [2]. Plastic, paper, and rubber components are the ones that contain the highest calorific values [21].

Modern incinerators can recover around 50 to 70% of the energy present in the US so 15 to 25% of this energy is transformed into electricity and the rest is transformed into thermal energy [23]. The relatively low electrical performance of the process reflects the limitation of the system operating at very high temperatures [2].

Although energy recovery is not the main objective of the incineration of UUs, this is an additional benefit, which helps maintain the viability of the operation, since the operational cost of the technique and also that of maintaining waste treatment is high [23], because in order to meet environmental legislation, incineration plants need to have more technical equipment to control air pollution, thus generating a higher cost [23].

On average, the technology can generate between 0.3 and 0.7 megawatt-hours (MWh) of electricity per ton of waste, depending on the size of the plant and the Lower Heating Value (LHV), i.e., the lowest waste heating value [18].

Incineration, like any conversion process, generates by-products. Solid emissions include ash and slag, which are non-combustible mineral parts, which are ferrous and non-ferrous alloys that can be extracted for recycling [22], and the rest of the ashes, similar to sand and gravel, are packed for a certain time to be used later on roads, buildings, or in the cover of landfills [4]. The ash resulting from combustion corresponds to 10% of the volume or 20 to 30% by mass of RSU [10].

Ash is an inorganic solid residue formed by mineral compounds and metal oxides. They can be subdivided into bottom grey or heavy gray, which are medium-sized powder materials that are not dragged by airflow, and fly ash or light ash, which is a particulate material with a thinner particle size that can be carried by combustion gases. Slag is a solid classified with higher granulometry and consists of the addition of non-combustible materials with products of the calcination of inorganic substances and ash sintering [15].

It is important to highlight that high temperatures (higher than 420 °C) are important to limit the formation of slag and fouling, which are accompanied by corrosion due to the presence of chlorine, mainly, and the accelerated wear of heat exchange surfaces [24].

In addition to solid particles, gaseous emissions are also generated by the process, such as sulfur oxides (SOx), carbon oxides (COx), nitrogen oxides (NOx), and hazardous metals, as well as carcinogenic emissions such as dioxins and furans, and polyaromatic hydrocarbons (HPAs), which are among the Persistent Organic Pollutants (POPs). Therefore, additional treatment is required in the combustion gas cleaning system before atmospheric emission [16,25].

The incineration process is not simple, so a disadvantage of this technique is that it has a high cost of implementation and operation. Another drawback is the potential unpleasant emissions of pollutants from incineration, as mentioned earlier, but these can be minimized with advanced technologies to control air pollution and segregate waste streams [26].

Although incineration is considered a path for sustainable waste management, it may not always be a viable disposal technique, as it depends largely on the characteristics of waste, which in turn are influenced by local demographics, social status and cultural differences, seasonal fluctuations, and topography [27]. Residues with high humidity and low calorific value can make the method unfeasible, as they decrease process efficiency. For this reason, it is very important that in the feasibility study of this process, the variable 'gravimetric composition' of the USR should be taken into account, since it influences the combustion power of the process.

However, the technique also has advantages, because it does not require large areas for installation, when compared to landfills, as the feeding of waste is continuous and drastically reduces the volume of waste, an advantage that is seen as the most important benefit of the incineration process.

Table 1 summarizes the main advantages and disadvantages of WtE technologies described in this subtopic.

**Table 1.** Advantages and disadvantages of energy generation technologies from waste.

| Burning Mass | |
| --- | --- |
| **Advantages** | **Disadvantages** |
| • They make it possible to process various types of waste [13].<br>• Reduction of volume and mass by 90% and 75%, respectively, without long periods of residence [10,15,22,24,28–30].<br>• There is enormous experience and international know-how in the face of a large number of plants in operation [13].<br>• Continuous operation that enhances scale gains [15].<br>• Energy use, especially when the residue (as received) has a lower calorific value (PCI) above 8000 kJ/kg (1911 kcal/kg) [31,32], which can be used in the form of water heating or transformed into electricity [10,29].<br>• Smaller area is required when compared to landfill disposal [15,24,28].<br>• Controlled incineration has less environmental impact than landfills [11] because it has lower greenhouse gas (GHG) emissions when compared to landfills [22,33]. | • Not feasible for small plants [33].<br>• High capital costs of the plant [24,28].<br>• Viability of the plant conditioned to processing capacity, usually above 6250 kg/h for RSU [15,31].<br>• The USRs have low energy content and high humidity, that is, relatively low heating value (LHV), especially in developing countries [33].<br>• The combustion of waste results in the formation of air pollutants (particulate matter, $SO_2$, HCl, HF, $NO_X$, CO, dioxins, furans, etc.) that require treatment to meet environmental legislation [15] and in the production of solid particles and metal-rich residues [11].<br>• Negative perception of the public strongly influenced by the emission of pollutants [13,34]. |
| **Gasification** | |
| • Application in small and medium scales [35].<br>• Possibility of using syngas in high-efficiency thermal devices (ICE and gas turbines or for biofuel synthesis) [18,35].<br>• Waste gasification has more favorable environmental results than incineration [18,24,35–37], as a limited form of dioxins, furans, nitrous oxides, sulfur oxides, and ash [24,38].<br>• Lower amount of secondary waste, which in some cases is produced in a less dangerous way, such as vitrified slag [24]. | • High operating cost [37].<br>• Need for pre-treatment to adjust moisture [37,39] and particle size [38].<br>• Still in the research phase [33].<br>• Not viable for large-scale commercial purposes [33]. |
| **Pyrolysis** | |
| • Reduction in the volume of waste from 70 to 90% [25].<br>• Shorter treatment time than in biological processes [25].<br>• Generation of more stable products, free of odor and pathogens [25]. | • Still in the research phase [33].<br>• Not viable for large-scale commercial purposes [33]. |

Source: prepared by the authors.

Through Table 1, it is noted that the mass-burning technology has greater economic viability to treat waste on a large scale. In this way, there is a greater number of plants in the world that use mass-burning technology instead of other techniques. However, there has recently been a greater interest in studying the other two techniques for the advantages they have, especially in environmental issues.

The next item will present the global scenario of Waste-to-Energy technologies.

### 2.2. Global Scenario

It is assumed that 1.2 billion tons of post-recycling MSW are generated annually in the world, but only 16.6% of it is treated using WtE technologies [13]. The International Renewable Energy Agency [40] estimates that the WtE recovery sector has enough capacity to generate 13 gigawatts (GW) of electricity on a global scale.

The first incinerator was built in 1875 in the city of London to satisfactorily carry out waste treatment, and there were already 121 incinerators in 1900 in England. However, the author states that it was only at the beginning of the 20th century that electricity started being produced from MSW incineration in Europe [41].

There has been a significant increase in European WtE plants only when the European Union introduced targets for diverting MSW from landfills to encourage energy recovery and recycling. In 1999, Directive 1999/31/CD [42] was proposed to reduce biodegradable waste disposal into landfills to minimize the production of methane and reduce global warming. In 2008, Directive 2008/98/EC [43] established a hierarchy of priorities for MSW disposal, in which waste-to-energy recovery takes greater priority over disposal into landfills. Figure 1 illustrates the hierarchy of these priorities in waste management.

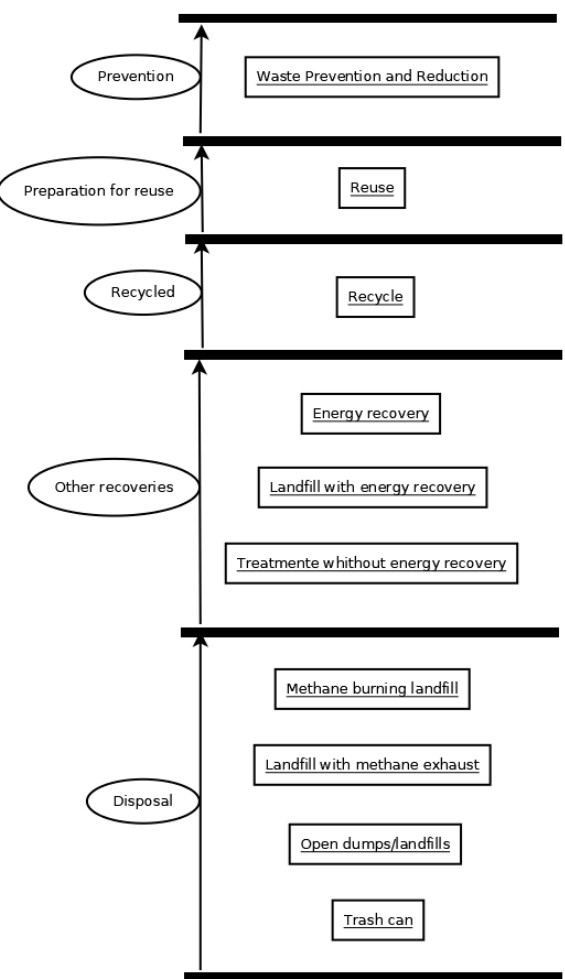

**Figure 1.** Waste priority hierarchy according to the European Union (Source: [4] adapted from the [17]).

Figure 1 reveals that there is great concern regarding waste minimization, since waste prevention, reuse, and recycling stand first in the hierarchy of priorities, followed by waste-to-energy recovery, and finally waste disposal into landfills with no energy recovery. It is worth mentioning that the Directive in question [43] emphasizes the importance of avoiding landfills as vehemently as possible, as well as recyclable material incineration.

Although these directives encourage the use of waste-to-energy recovery instead of landfill disposal, Directive 2000/76/EC [44] is forceful in establishing that all WtE plants must comply with strict atmospheric emission standards through waste collection, constant monitoring, and proper treatment.

Between 1995 and 2012, there was a 42% reduction in the amount of MSW dumped into sanitary landfills as a result of the encouragement of waste-to-energy recovery in Europe. On the other hand, the amount of waste recovered in WtE plants increased by 80%. In 2012, energy was recovered in 456 WtE plants across Europe, which prevented 79 million tons of solid waste from being disposed of in landfills. In 2015, 90 million tons of waste were treated by WtE plants, which supplied 18 million inhabitants with electricity and 15.2 million inhabitants with heat. In 2016, the number of WtE plants in operation rose to 514 units which processed 263,314 tons of MSW daily [4].

Currently, 10% of district heating across Europe comes from WtE plants. In cities like Brescia, Malmö, or Klaipėda, heating from these plants covers 50% or more of the heating demand. Regarding electricity generation, around 19 million people a year are supplied by WtE plants in Europe [45].

Figure 2 shows the trend of MSW treatment in 2019.

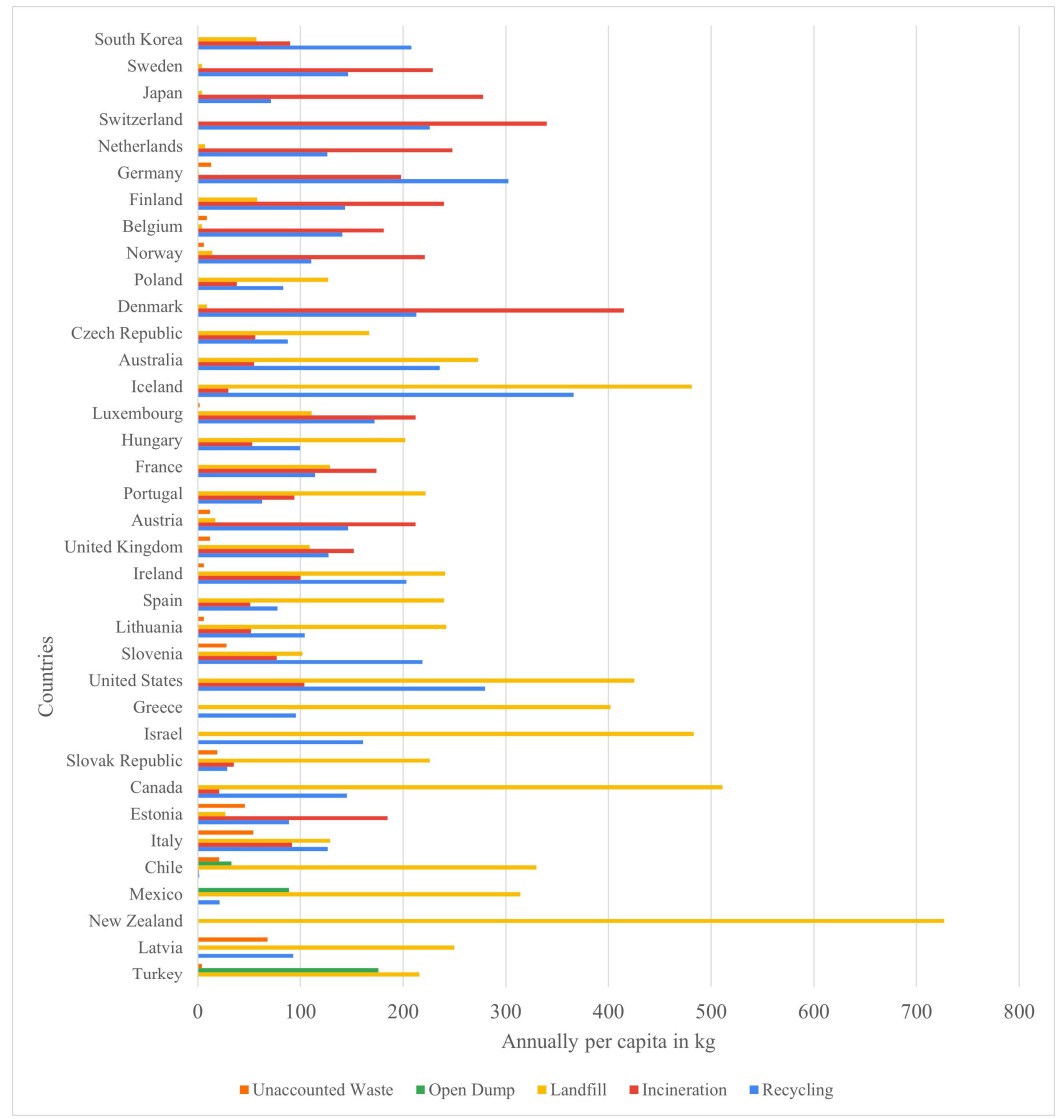

**Figure 2.** Municipal waste treatment in 2019 (Source: [46]).

It can be concluded from Figure 2 that in several countries around the world, incineration stands out compared to other sources of disposal, as is the case in some European countries, such as Germany, Denmark, Switzerland, Holland, and Sweden, among other countries, as well as in Japan. It can be concluded that the countries that most use this technology are countries with a smaller territorial area and this may be due to the lack of space for landfilling their waste.

Countries having the highest recycling rate are also those with the highest waste-to-energy recovery, thus reducing the use of landfills to nearly zero, as is the case of the first eight countries shown in Figure 2. In Brazil, there is a discreet rejection of the application of WtE technology due to considerations that this would affect recycling cooperatives, but this objection is much more due to social, political, and economic conflicts in the country.

As future goals, Europe aims to accomplish the following: (a) expand the capacity and number of WtE plants to process over 40 million tons of MSW yearly; (b) reduce MSW disposal into sanitary landfills from 25% to 10 % until 2035; and (c) increase waste recycling by up to 65% by 2035. Through this policy, the European Union will (a) generate 18 TWh of energy, either in the form of heat or electricity, and (b) reduce 115 million tons of $CO_2$ generation aiming to lower greenhouse gas (GHG) emissions into the atmosphere [47].

Considering the data pointed out in Figure 2, and deepening the observations regarding the existence of WtE plants, we can highlight that in Japan, the large projects of WtE plants were conceived in the 1960s with the initial aim of increasing land availability due to its high cost in the country and the concern about the quality and scarcity of water, as well as economic and environmental benefits of improving the technique's effectiveness, which was posteriorly taken into account [4]. Currently, over 80% of MSW is incinerated and 20% is recycled in Brazil. From this percentage, 24.5% recovers energy at a conversion rate of approximately 200 kWh per ton of MSW. In Tokyo, the electricity conversion rate of these incinerators reaches up to 390 kWh per ton of MSW. In Kobe, 16.2% of its electricity demand and 25% of its hot water demand are supplied through incineration at an average conversion rate of 300 kWh per ton of MSW [48–50].

In South Korea, there are 35 WtE plants in operation, which incinerate 25.02% of all MSW produced countrywide. It should be noted that these plants have excellent performance, as they have very low levels of pollutant emissions [4,13].

China will have had 339 WtE plants in operation with an installed capacity of 7.3 GW of electricity by the end of 2017, thus being considered the country whose WtE plants have the largest installed capacity worldwide. As a reference, such installed capacity is equivalent to 40% of that of all countries belonging to the Organization for Economic Co-operation and Development (OECD) altogether. China's Five-Year Plan estimates that over 13 GW of installed power capacity will have been reached by 2023, which corresponds to the same installed power capacity as that of the Itaipu Power Plant, which satisfies 15% of the Brazilian electricity demand. Finally, it is estimated that the country's WtE plants will have been able to process 260 million tons of MSW by 2025 [4,51].

In New York, some public incineration plants started to be built incipiently after 1906. After 1950, with land value appreciation, local governments started building waste incineration chambers and used smoke to purify water. Modern charcoal filtration systems to minimize hazardous particulate emissions were only developed after the United States Environmental Protection Agency (USEPA) laboratory had identified the presence of atmospheric dioxins and other toxic substances in 1977 [52].

According to the Solid Waste Association of North America (SWANA) [53], the United States encountered several obstacles that eventually hampered the process of developing the WtE industry, among which are strict conditions for controlling atmospheric emissions, lack of proper disposal of ashes which in turn have increased the operating costs of plants, the fact that sanitary landfills are cheaper options for final MSW disposal, and the obstacles posed by the electric energy industry hindering the sale of electricity from WtE plants.

Currently, the United States' MSW is treated as follows: 26% is recycled, 9% is composted, 52% is sent to landfills, and 13% to WtE plants [4]. According to data from the

United States Environmental Protection Agency (EPA) [54], in Brazil there are 86 facilities in the country recovering energy through MSW incineration, with a capacity to process 25 million tons of waste yearly, thus generating 2720 MW of electricity.

There are 5 WtE plants in operation in Canada, but only one of which uses incineration technology, which has the total capacity to treat 2272 tons daily, i.e., 3% of all MSW generated in the country [4].

India has 8 WtE plants in operation, totaling 94.1 MW of installed power. Also, according to the same author, 50 more plants are being built with enough capacity to process 30,000 tons of MSW a day, which totals 398 MW of installed capacity [55].

In Russia, there are only 4 WtE plants (located in Moscow) with enough capacity to process 1179 tons of MSW a day [4].

Latin America faces a major problem regarding waste management, especially concerning poor environmental governance and, consequently, Integrated Waste Management practices. Due to low financial investments in the sector, hiring more workers and investing in new technologies is impractical [56]. Moreover, [56] states that Latin America is also lagging regarding WtE plants. However, the first plant is going to be built in Mexico City whose daily processing capacity is 4500 tons of MSW. This plant is regarded as the plant having the largest installed capacity (110 MW) in the world to date. Its electricity production is going to be sold to the Mexico City Subway System at R$ 414.00/MWh based on the collection of a tipping fee of R$ 80.00 (US$ 15.47—07/31/2022) per ton of waste [4].

In Brazil, there were some experiments on incinerators (Table 2), but these have no pollution control system or are incapable of estimating energy recovery. Given such a lack of equipment to control emissions, especially of dioxins, furans, and other toxic substances, incineration plants were banned.

Currently, a few small pyrolysis and gasification plants are operating on Refuse Derived Fuel (RDF) in Brazil, as is the case of a plant located in the municipality of Mafra, Santa Catarina. In Boa Esperança, Minas Gerais, construction began on a plant consisting of an MSW processing unit, a gasification unit, and a generation unit. This plant will process 30 tons per day of CDR [57].

**Table 2.** Some Brazilian experiments on incinerators.

| Brazilian Cities | Implementation | Completion |
|---|---|---|
| Manaus (AM) | 1896 | 1958 |
| Belém (PA) | Early 20th century | 1978 |
| Araçá (SP) | 1913 | 1948 |
| São Paulo—Pinheiros (SP) | 1949 | 1990 |

Source: Adapted from [58].

There are also several studies [57,59] and pilot plants [60] that study the viability of the gasification method using MSW, mainly for small and medium-sized municipalities, since 97% of Brazilian municipalities have less than 200,000 inhabitants [60].

There are still prospects for the construction of two other Waste-to-Energy Recovery (WtE) Plants using the mass-burning technology, both in the state of São Paulo, which is going to be better described in Section 2.3. [4].

In addition to these plants that have already been through or are currently undergoing a licensing process, there are also studies carried out by Brazilian researchers on the possibility of implementing WtE plants in different contexts of the country. One study pointed out that incineration can generate enough electricity to supply 39% of local residences in the city of Campinas, while biodigesters are capable of generating enough electricity to supply 1% of its residences, as long as MSW plants keep using 20% of recyclable materials available [10].

In a work on Isolated Systems covering a large region of the state of Amazonas, the author points out that, for such a scenario, waste-to-energy recovery through pyrolysis would be attractive for larger municipalities and the use of gasification for small commu-

nities would be interesting since these remote places are not connected to the National Interconnected System (SIN) and are thus dependent on thermoelectric generation from fossil fuels, which are highly polluting and have elevated costs [61].

Another Brazilian case study on WtE recovery from MSW was conducted in the city of Santo André, state of São Paulo [62]. The authors found that the electricity generated from the city's MSW would be enough to supply 8.89% of the population with electricity; thus, it would be possible to diversify its energy matrix.

In another study, different scenarios were considered for MSW treatment in Varginha, state of Minas Gerais. Through this research, the authors showed that only 150 kW of electrical energy recovery is economically viable, while the greatest environmental benefits in terms of gas emissions and energy recovery were found in a scenario consisting of energy recovery through recyclable materials, anaerobic digestion, and incineration used simultaneously [34].

Thus, WtE recovery from MSW is still incipient in Brazil, although studies and some initiatives are being developed to show that it can be a solution to some scenarios in the country, in addition to the fact that it can also be associated with other alternatives to reap greater economic, environmental, and even social gains.

### 2.3. Perspectives of Mass-Burning Technology in Brazil

In Brazil, there is still no WtE recovery from MSW. The National Electric Energy Agency (ANEEL) through the Generation Information Bank (BIG) describes the existence of a WtE plant (UTE Tremembé) in its Energy Matrix aimed at MSW heat recovery with an installed capacity of 4.27 MW; however, this classification is wrong, as it is a landfill gas recovery plant [4].

Plants using biogas from MSW landfills are already more advanced in Brazil where there is a total of 22 plants with an installed capacity of 164.32 MW, representing 0.09% of the Brazilian energy matrix [63]. However, although the capture of methane from landfills can generate electricity, it is not an efficient mechanism when compared to WtE plants, especially by taking mass-burning technologies into account. A mass-burning plant's energy efficiency is ten times greater than that of landfill gas plants, and a mass-burning plant produces an average of 600 kWh of electricity per ton of MSW, whilst landfill plants generate 65 kWh per ton. Furthermore, electricity is generated slowly over time in landfills due to longer biogas extraction rates, while electricity is generated instantaneously in mass-burning plants [52].

Waste-to-energy recovery through mass burning can be an alternative treatment for some locations in Brazil by charging tipping fees and tariffs based on the charging rates defined by the National Sanitation Policy [64], which has been recently amended by the New Regulatory Framework for Sanitation [65]. However, Brazilians are not used to paying for such services, unlike in other countries, which creates an adverse reaction from the population, together with greater concern about atmospheric emissions generated as a result [41]. Therefore, the implementation of mass-burning plants for specific contexts in Brazil becomes a major challenge for urban cleaning management.

Another alternative for the implementation of energy recovery through waste, offered by the PNRS [66], is the regionalization of MSW disposal, so that technical and political efforts can be combined to optimize waste management. The update of the New Legal Framework for Sanitation [65] also reinforces the idea of regionalized solutions.

There are already cases in Brazil in which regionalization and the formation of municipal consortia have worked for the grounding of waste [67]. In this way, for mass-burning technology, municipal consortia could also be an alternative, as it would help in the provision of services in addition to bringing economic, administrative, and environmental advantages to municipalities.

The main regulatory frameworks on waste-to-energy recovery of MSW in Brazil in the state of São Paulo, Brazil's energy capacity to generate electricity from MSW, and the main WtE plants that are being implemented in Brazil are going to be presented as follows.

2.3.1. Regulatory Frameworks

The most significant legal instrument supporting integrated solid waste management is the Brazilian National Policy on Solid Waste (PNRS), which has been established through Federal Law No. 12.305 [66]. The Law governs, among forms of environmentally friendly disposal, energy recovery and use, provided that there is proven technical and environmental feasibility, including the monitoring of gas emissions approved by the environmental agency.

The PNRS was formulated in the same year, and Decree No. 7404 [68] was approved to govern it because there must be legislation on waste management in the country. In this Decree, articles 36 and 37 establish that waste-to-energy recovery must comply with regulations introduced by respective agencies and that they should be operated in conjunction with the Ministry of the Environment, Mines, and Energy and Cities.

However, the National Energy Policy was formulated in 1997 before the enforcement of the PNRS and the Decree to govern solid waste management, which has been enacted by Law No. 9.478 [69] governing the use of alternative energy sources, which comprises waste-to-energy recovery from MSW.

In 1999, CONAMA Resolution No. 264 [70] (p. 1) was introduced, aimed at "licensing rotary kilns for clinker production towards waste co-processing activities". Subsequently, in 2002, CONAMA Resolution No. 316 [5] establishes the procedures and criteria for operating waste heat recovery systems. It is worth mentioning that it emphasizes the importance of heat recovery over the implementation of a prior segregation program for recycling or reuse purposes. However, it requires less stringent rates if compared to international standards of gas emissions resulting from waste mass burning regarding the licensing and operation of WtE plants.

For comparison purposes, the emission limits for dioxins and furans are 0.1 ng/Nm$^3$ in the European Union, 0.13 ng/Nm$^3$ in the United States and 0.50 ng/Nm$^3$ in Brazil. For particulate matter the limits are 10 mg/Nm$^3$ (UE), 20 mg/Nm$^3$ (USA) and 70 mg/Nm$^3$ (Brazil). Limits for vaporous and gaseous organic substances only exist in the EU (10 mg/Nm$^3$) [29]. Regarding emission limits of hydrogen chloride (HCl), the European Union (10 mg/Nm$^3$) is stricter than the United States (29 mg/Nm$^3$) and Brazil (80 mg/Nm$^3$). As for Hydrofluoric Acid (HF) emissions, the limits only appear in Brazilian (5 mg/Nm$^3$) and EU legislation (1 mg/Nm$^3$) [29].

For sulfur dioxide ($SO_2$) emissions, Brazil appears to be more permissive (280 mg/Nm$^3$) than other countries (85 mg/Nm$^3$ in USA and 50 mg/Nm$^3$ in UE). Finally, for nitrogen monoxide (NO) and nitrogen dioxide ($NO_2$), the emission limits are, respectively, 200 mg/Nm$^3$ for the EU, 305 mg/Nm$^3$ for the USA and 560 mg/Nm$^3$ for Brazil [29].

The data presented above that make a comparison between the emission limits of various pollutants established by the European Union, the United States, and Brazil, shows that Brazilian legislation, still valid in the country, is less restrictive concerning gas emissions from waste mass burning when compared to the United States and the European Union. This must be reviewed since there are incentives for the implementation of WtE plants in Brazil, mainly in its metropolitan regions. Thus, less strict legislation can lead to public health problems, as well as to impacts on the environment.

Given that there has been an increase in air pollution over the years due to economic growth in the 2000s in Brazil, there was a need to establish limits on pollutant emissions. Therefore, CONAMA Resolution No. 382 [71] was formulated in 2006, and CONAMA Resolution No. 436 [72] in 2011, through which maximum limits for atmospheric emissions referring to fixed sources were established. Their difference lies in the plant's license date of issue, as CONAMA Resolution n$^o$ 436 lays out the ones that have been granted or those whose license requests were made before 2 January 2007.

Another important framework was the formulation of the Brazilian Basic Policy on Sanitation, Law No. 11,445 [64] in 2007. This legislation ratified that collection and environmentally friendly solid waste disposal are also sanitation activities as well as urban

drainage, in addition to trivial activities of sewage collection and treatment, and drinking water treatment and distribution.

From an environmental standpoint, the National Policy on Climate Change (PNMC), Law No. 12,187 [73] has made an important contribution to the theme, since it proposes the development of technologies to minimize Greenhouse Gas (GHG) emissions, and waste-to-energy recovery from MSW mass burning are included in this category, as it reduces direct emissions of methane into the atmosphere due to a previous gas treatment.

In the following year, Decree No. 7390 [74] was enacted aiming to govern the PNMC, which was later revoked by Decree No. 9578 [75]. Its main contribution to the theme was to determine a baseline for GHG emissions in 2020.

In 2019, Interministerial Ordinance No. 274 [76] was formulated to regulate energy recovery (heat recovery), particularly from MSW, i.e., the implementation and operation of WtE plants. However, according to [4], it did not address the matter to the same extent that European Directives do.

The Brazilian Association of Technical Standards (ABNT) defined Brazilian Standard (NBR) No. 16,849 [77] in 2020, which establishes the requirements for waste-to-energy recovery from municipal solid waste, either with or without the incorporation of other class II waste—non-hazardous. This standard is very important for waste-to-energy recovery from MSW to be carried out rationally, i.e., using appropriate technologies and promoting safe practices.

In the same year, the New Legal Framework for Sanitation, Law No 14.026 [65], was enacted which encouraged the provision of services together with the private sector to overcome the deficit in the sanitation sector and universalize services. Although the premise of universalization of services is true and important, there are, on the other hand, criticisms regarding service privatization, especially concerning tariff increases from the end of a cross-subsidy, in which the damage costs in small municipalities are covered by the profit reaped from the most populated areas.

Ultimately, CONAMA Resolution nº 499 was issued in 2020 [78] to license waste co-processing in cement kilns, which was already governed by CONAMA Resolution nº 264 [70]. CONAMA Resolution nº 499 [78] does not update the one issued in 1999, but it is a setback since it revokes several devices ensuring slightly greater environmental safety [79].

Figure 3 depicts a timeline presenting the main frameworks mentioned previously for energy recovery from MSW in Brazil.

In the state of São Paulo, there are also some legal instruments related to the theme. In 1992, the State Policy on Basic Sanitation was formulated, Law No. 7,750 [80], which aimed to govern sanitation services in the state of São Paulo, from planning to actions to be taken. It states that municipalities are responsible for managing municipal sanitation facilities and services, including final waste disposal.

The Environmental Company of São Paulo State (CETESB) defined Technical Standard P4-263 aiming to establish procedures for the use of waste in clinker furnaces in 2003 [81]. Thus, it ensured another form of final waste disposal that is also considered environmentally friendly within the criteria established by the Standard.

In 2006, the São Paulo State Policy on Solid Waste Management (PERS) was enacted, Law nº 12,300 [82], in which, among enacted provisions, the integrated and shared planning of waste management and the incentive for its implementation are worth highlighting, as well as research on clean technologies aimed at processes of treatment and final disposal of MSW.

Another important instrument was the creation of the State Policy on Climate Change (PEMC), Law nº 13,798 [83], in which one of its main purposes is to incorporate a larger share of renewable resources into the energy matrix.

Decree No. 55,947 [84] was issued to govern the PEMC in the following year. Its main contribution to the theme was the establishment of energy recovery from waste as a way to face the effects of climate change.

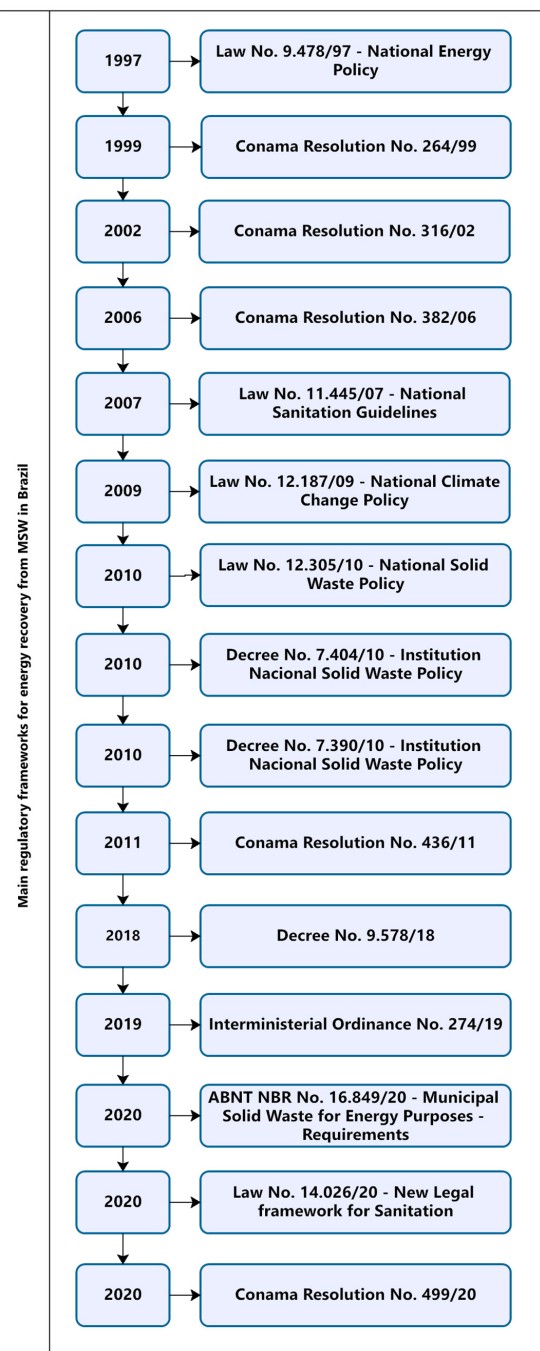

**Figure 3.** Main regulatory frameworks for energy recovery from MSW in Brazil (Source: prepared by the authors based on [5,64–78]).

In 2009, Resolution SMA n° 79 [6] was formulated, which was based on regulatory practices of Directive 2000/76/EC [44] to ensure that pollutant emissions from MSW mass burning must be inspected. In 2013, new air quality standards were established by State Decree No. 59,113 [85] and such pollutants that were not considered in this legislation are going to be subject to regulation based on recommendations from the World Health Organization (WHO).

It is also worth mentioning two Board Decisions by the Environmental Company of São Paulo State (CETESB) which are regulatory frameworks for WtE plants. The first CETESB Board Decision No. 326, 2014 [86], provides the criteria for verifying compliance with the emission limits of parameters established by Resolution SMA/SP No. 79, 2009 [6], concerning the licensing of municipal solid waste heat treatment activities in WtE plants [60].

The second CETESB Board Decision No. 034, 2015 [87], establishes technical standards for Human Health Risk Assessment due to unintentional exposure to atmospheric emissions of dioxins and furans that condition the issuing of a prior environmental license for WtE recovery plants.

Lastly, Resolution No. 38 of 31 May 2017 [88] (p. 1), by the Department of Infrastructure and Environment (SIMA), established "guidelines and conditions for licensing and operation of activities of energy recovery from the use of Refuse Derived Fuel from MSW—URDF, in Clinker Production Furnaces" and Resolution No. 47 [89] put forward by the same Department established "guidelines and conditions for the licensing of units for the preparation of Refuse Derived Fuel from Solid Waste—RDF and energy recovery from the use of Refuse Derived Fuel—RDF" on August 6, 2020.

Figure 4 presents a summary of the main regulatory frameworks mentioned above for the state of São Paulo through a timeline.

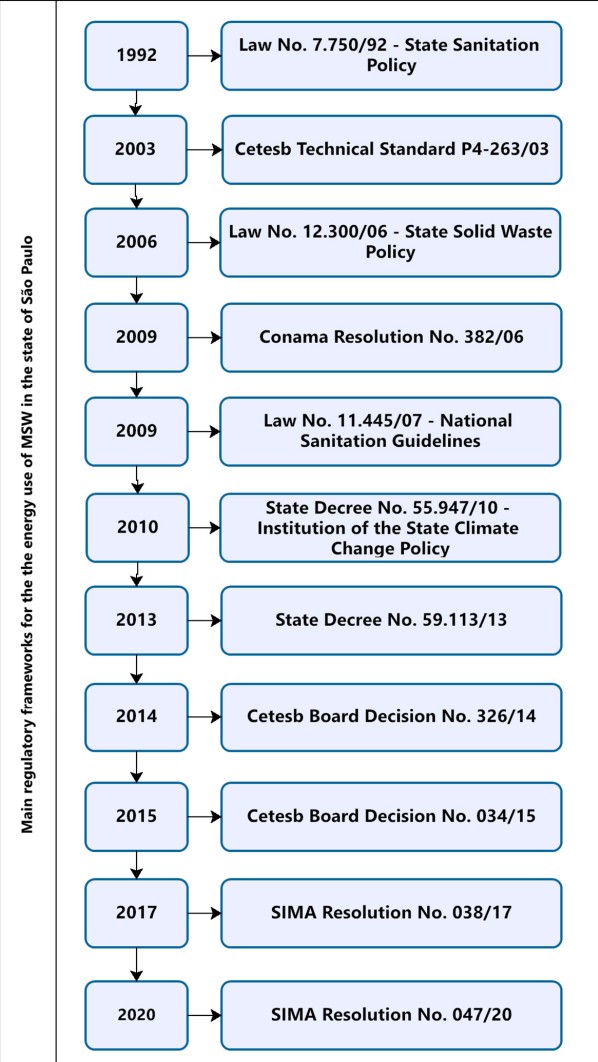

**Figure 4.** Main regulatory frameworks for the energy use of MSW in the state of São Paulo (Source: prepared by the authors based in [6,80–89]).

It should be noted that energy recovery from MSW is not just limited to these instruments. The purpose herein is to point out some important frameworks for initial contact with the theme.

### 2.3.2. Brazilian Energy Capacity

As pointed out by the Brazilian Energy Balance [90], the country has enormous diversity in its energy matrix (Figure 5) and renewable energy sources (hydraulic, biomass, wind, and solar), accounting for 84.8% of the Brazilian electrical matrix (Figure 6).

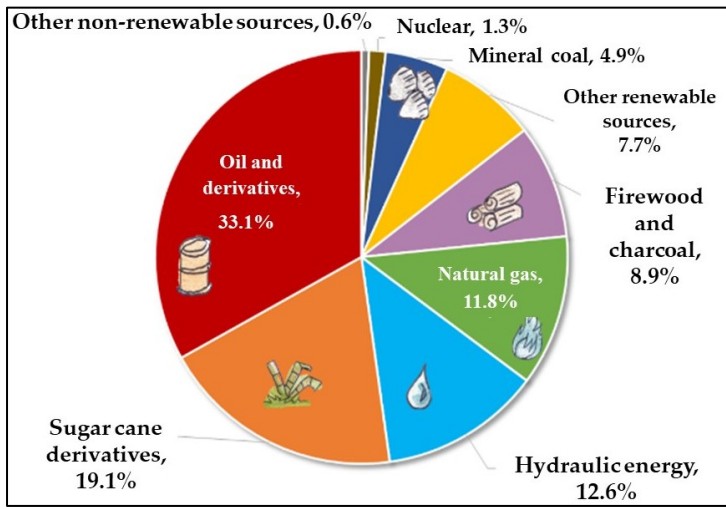

**Figure 5.** 2020 Brazilian Energy Matrix (Source: translated from [90]).

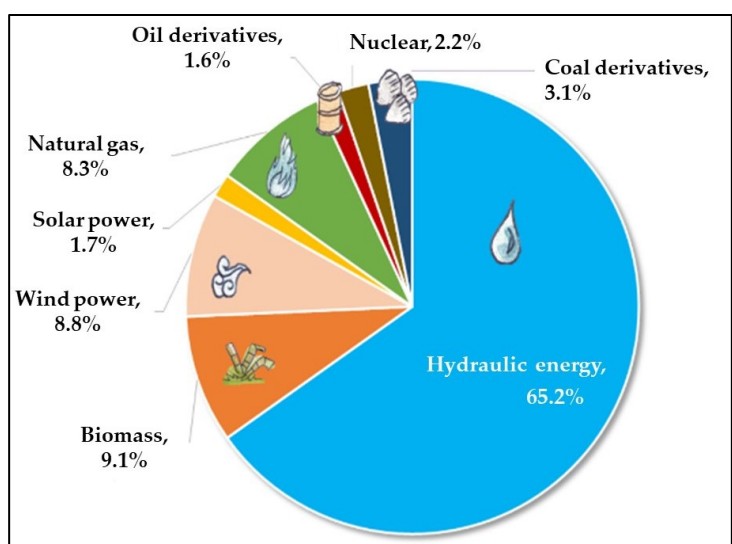

**Figure 6.** 2020 Brazilian Electrical Matrix (Source: translated from [90]).

It can be observed that Figure 6 is an expansion of Figure 5; however, Figure 6 highlights the subdivision specifically of the sources that are transformed into electrical energy. From Figure 6, it can also be seen that the electricity supply from MSW is not listed in Brazil's energy matrix, since the country, in fact, still does not generate electricity from such a source.

The efficiency of WtE plants in the conversion to electrical energy is relatively low, between 20 and 25%, if the entire amount of MSW in Brazil (approximately 192,000 tons per day) were incinerated, it would be possible to generate 35 terawatt hours (TWh) a year, considering an average of 0.5 MWh per ton. In a more optimistic view, this value could reach 50 TWh/year, with 700 kilowatt-hours (kWh) per ton [2]. If 80% of the country's MSW were incinerated, with an efficiency of around 30%, the electricity generated would be in the order of 2902.6 GWh/month, which would be enough to supply almost 25% of Brazilian homes [29].

However, considering a very high percentage of MSW incineration is harmful to sustainable management, a share of this waste might be recycled or composted. Recovering 35% of the country's MSW through WtE plants would be ideal, as approximately 1300 GWh/month could be generated, i.e., an amount capable of supplying 3.29% of the country's electricity demand, thus providing greater diversity to its energy matrix [4].

The most limiting issue for the development of energy recovery from MSW in Brazil is the fact that the country's energy matrix is predominantly from renewable sources, especially when compared to Europe and Asia [91,92]. Thus, for the Brazilian electricity sector, energy generation from this source seems to offer no significant environmental contributions when compared to existing sources and those that are highlighted due to their efficiency and sustainability, as is the case of photovoltaic energy. However, waste management can be an important form of treatment and final disposal in some locations of the country, especially in highly populated regions where waste landfilling becomes increasingly complex due to the scarcity of appropriate areas for disposal.

Therefore, energy recovery from MSW cannot be disregarded in the Brazilian context, both in terms of its contribution to waste management, as well as to the energy sector which, although it is not considered environmentally significant, is important for diversifying the country's energy matrix, thus minimizing the degree of energy dependence and improving energy safety.

### 2.3.3. Waste-to-Energy Recovery Plants (WtE)

A Waste-to-Energy Recovery Plant (URE) can be defined as "one aimed to recover energy using heat energy generated by waste mass burning" [77] (p. 7).

In Brazil, there are two well-established mass-burning WtE projects that have already undergone the environmental licensing process, both in the state of São Paulo.

The first one is located in Barueri, where investments of around R$ 320 million are planned for the recovery of 825 tons of MSW per day with the capacity to generate 17 MW of electricity. Such a WtE plant was designed to receive waste from the cities of Barueri, Carapicuíba, and Santana de Parnaíba with 30 years of estimated lifespan, considering an operation of 8000 hours per year [93,94].

The other project is located in the municipality of Mauá, where the WtE plant will be implemented on the property of Lara Residue Treatment Center Ltd., which already houses the Lara Sanitary Landfill. The plant will occupy an area of 72,025 m$^2$ with the capacity to treat 3000 tons of MSW per day and an installed capacity of 77 MW. The WtE plant will treat MSW from the municipalities of Mauá, Diadema, Ferraz de Vasconcelos, Itanhaém, Juquiá, Ribeirão Pires, Rio Grande da Serra, São Bernardo do Campo, and São Caetano do Sul [95].

It is worth mentioning that the technologies used by both WtE plants can be considered a Clean Development Mechanism (CDM), as emissions generated by the process are treated before being released, thus minimizing the risk of contributing to climate change. However, some researchers question WtE plants from the Life Cycle Analysis (LCA) point of view, since they consider that WtE plants can discourage the reuse and recycling of materials, as the material undergoes a linear process ("cradle to grave") and not circular ("cradle to cradle") [96]. Thus, priority should be given in these projects to only the use of non-recyclable waste as raw material for the process and not any types of waste.

In addition to environmental issues, risks to human health were also taken into account in the licensing process. The levels to which the local population will be exposed are within the tolerable limits established by federal and state laws, and by regulations specific to the subject.

Thus, for large urban centers, as is the case of the Metropolitan Region of São Paulo, where there is continued population growth and a consequent increase in waste generation, there is a mounting challenge of finding areas available for landfills; therefore, WtE plants are possible solutions for final waste disposal in an environmentally friendly manner.

## 3. Final Considerations

This article allowed contextualizing Waste-to-Energy recovery technologies in a global context, especially concerning MSW mass-burning technologies for the Brazilian scenario, through collecting technical information from scientific articles and current legislation on the subject, as well as for the circumstances found for the Brazilian WtE plants.

It is noted that the MSW mass-burning technique has been gaining more and more notoriety worldwide since new technologies used in the process make it more efficient and capable of meeting environmental standards.

To universalize basic sanitation throughout Brazil, the New Legal Framework for Sanitation [65] is aimed to stimulate the provision of services in the area of waste management as well, thus arousing the interest of companies in the WtE sector to invest in mass burning for some Brazilian contexts, especially in populous regions. Despite the tendency of this technique to be spread out, it faces challenges in its incorporation, mainly due to its high cost of implementation. However, it cannot be overlooked for populous countries like Brazil, since its economic benefits tend to outweigh those of final disposal carried out in sanitary landfills in the long term.

Another issue worth being observed is the environmental sustainability generated by the technique when compared to sanitary landfills, as it allows greater energy recovery, in addition to releasing less GHG into the atmosphere.

Another point worthy of attention on behalf of authorities is the fact that Brazilian legislation on emission standards is less restrictive than in the United States and Europe and, over the years, instead of ensuring greater environmental safety, regulations are being relaxed, which could trigger greater public health concerns and, therefore, must be reviewed.

Ultimately, it is worth noting the great contribution of this energy recovery technique in the Brazilian context. As a developing country, Brazil needs to diversify its energy matrix to guarantee its energy independence and, therefore, have greater energy safety for the advancement of industrialization, particularly concerning agribusiness in the country, since this sector leverages the Brazilian GDP. Therefore, the private sector and governments must incentivize the debate on other options for waste management and encourage a continuous improvement of these energy recovery technologies by taking into account environmental and public health safety.

**Author Contributions:** Conceptualization, N.D., M.C.R. and F.A.d.S.; methodology, N.D. and M.M.N.; software, C.P.C. and L.R.A.G.F.; validation, C.P.C. and L.R.A.G.F.; formal analysis, M.M.N.; investigation, N.D.; resources, M.C.R.; data curation, M.M.N.; writing—original draft preparation, N.D.; writing—review and editing, N.D., M.C.R. and M.C.R.; visualization, F.A.d.S.; supervision, C.P.C.; project administration, L.R.A.G.F.; funding acquisition, N.D., M.M.N. and L.R.A.G.F. All authors have read and agreed to the published version of the manuscript.

**Funding:** This research was funded by Personnel Improvement Coordination of High Level (CAPES), grant number 88887.490160/2020-00 (ND) and by the National Council for Scientific and Technological Development (CNPq) for the research productivity grants awarded (Process #313339/2019-8 (MMN) and #315228/2020-2 (LRA)).

**Acknowledgments:** The authors wish to acknowledge the Postgraduate Program in Agribusiness and Development (PGAD) of the School of Sciences and Engineering of São Paulo State University (UNESP), the Brazilian National Council for Scientific and Technological Development (CNPq), and also the Personnel Improvement Coordination of High Level (CAPES) for financial support.

**Conflicts of Interest:** The authors declare no conflict of interest.

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
