# Peer review of "Waste-to-Energy Recovery from Municipal Solid Waste: Global Scenario and Prospects of Mass Burning Technology in Brazil"

_sustainability, doi:10.3390/su15065397_

Round 1
Reviewer 1 Report
The manuscript describes the global scenario of waste-to-energy recovery from municipal solid waste and outlines the potential of the development of mass burning technology in Brazil.
The topic is of scientific research and the manuscript is well structured. As a general comment, too much space is given to the description of the story of the regulatory frameworks (Section 2.2.1), which is not effectively summarized with the aim of conveying key and useful information to understand the future steps of the technology. Moreover, I would add a Section after the Introduction to generally present the different technologies used in waste-to-energy recovery (e.g., incineration, gasification etc.), highlighting technological aspects, limits and potentialities of each of them. Subsequently, I suggest focusing on incineration plants (explaining why they have a more promising future compared to the other technologies) and describe (for example in a Subsection) their main components along with the last technological advances.
The authors should also consider the following more specific comments:
1) Pag. 3, lines 126-128: it is not clear the reason why WtE technology is not used in Brazil. If WtE prevails on recycling, its use should be favored, but it seems that the sentence says the opposite. Please reformulate it.
2) Fig. 2: since Fig. 2 is in Section 2.1 (Global scenario) I would have expected different diagrams showing the trend of MSW treatment at global level and not only in Europe. Moreover, more recent data (beyond 2017) should be displayed.
3) Pag. 5, lines 180-181: it is said that in Brazil 13% of MSW is treated by WtE plants. This is a discrepancy with the several statements throughout the paper saying that, at present, no WtE plants are installed in Brazil (see e.g., line 250).
4) Are the emission limits for Brazil shown in Table 2 still valid today? If yes, please specify it clearly when you present it.
5) The difference between Fig. 5 and Fig. 6 should be better explained in the text. Moreover, captions of the two figures should be different to explain this difference.
6) Pag. 11, lines 437-442: it should be specified which efficiency values have been assumed for these plants.
Author Response
Dear reviewer,
All authors are grateful for their notes and comments. We consider all of them very important for the improvement of our manuscript.
We understand all placements and strive for full compliance or justification.
Next, we present the answers corresponding to the changes made to the manuscript.
Considering your general comment, “too much space is given to the description of the story of the regulatory frameworks (Section 2.2.1), which is not effectively summarized with the aim of conveying key and useful information to understand the future steps of the technology”, we would like to justify our choice to keep the text of Section 2.2.1 because we consider this to be an excellent summary for anyone studying WtE in Brazil and it may certainly need to see which legislations apply to that context. Thus, we consider that, depending on your position, the objective of transmitting important and useful information to understand the future steps of the technology, in case it is implemented in the future, can be met. However, if you still consider that the topic should be removed or summarized, we would appreciate your suggestions.
Next, we fully agree with your considerations about adding a Section after the Introduction to present in general the different technologies used in the recovery of waste for energy, for example, incineration, gasification, etc., highlighting technological aspects, limits, and potential of each one of them, including seeking to focus on incineration plants, explaining why they have a more promising future compared to other technologies and describing their main components along with the latest technological advances. Thus, to better respond to this request, we authors decided to insert subtopic 2.1 (between lines 76 to 185).
In response to your more specific comments, we have a list of our actions to respond to:
“1) Pag. 3, lines 126-128: it is not clear the reason why WtE technology is not used in Brazil. If WtE prevails on recycling, its use should be favored, but it seems that the sentence says the opposite. Please reformulate it.”
For this specific case, our actions were guided to reformulate the sentence to give a better understanding of what we meant (lines 223-244).
“2) Fig. 2: since Fig. 2 is in Section 2.1 (Global scenario) I would have expected different diagrams showing the trend of MSW treatment at global level and not only in Europe. Moreover, more recent data (beyond 2017) should be displayed.”
In response to this second specific question, we inform you that the European data have been updated, including the change in Figure 2, which gained new information (lines 223-244).
“3) Pag. 5, lines 180-181: it is said that in Brazil 13% of MSW is treated by WtE plants. This is a discrepancy with the several statements throughout the paper saying that, at present, no WtE plants are installed in Brazil (see e.g., line 250).”
At this point, we recognize the error, and in this case, the text does not refer to Brazil, but to the USA (since there is a connection with the previous paragraph). The mistake was made by the translator when translating the word “country”, he replaced Brazil with the USA. But this has since been corrected (line 286).
“4) Are the emission limits for Brazil shown in Table 2 still valid today? If yes, please specify it clearly when you present it.”
According to your question, we replied that yes, the indicated limits are still valid in Brazil. To resolve the issue, a text was inserted that guaranteed this condition (line 425)
“5) The difference between Fig. 5 and Fig. 6 should be better explained in the text. Moreover, captions of the two figures should be different to explain this difference.”
In response to your statement about the difference between the two Figures, we have to say that a greater differentiation between the two was made in the manuscript, including a change in the caption (in which there was also a translator error, for which we apologize again) the text is found between the lines: 535, 542, 544-546).
“6) Pag. 11, lines 437- 442: it should be specified which efficiency values have been assumed for these plants.”
In response to your request, efficiency values have been entered (lines: 549-550, 554).
Thus, considering the changes made, we hope to have met your requests and we await your approval, or further comments, for the approval of the manuscript.
Yours sincerely,

Reviewer 2 Report
Authors discusses mainly incineration processes. In fact, this is a process already in place in several industrialized countries and the Brazilian environmental legislation addresses is in an adequate way.
However, authors fail when discussing economic feasibility aspects. It does not seem incineration process is feasible for plants smaller than 600 t/d of MSW. This corresponds in average for municipalities with around 600,000 inhabitants (considering an average of 1.0 kg/hab/day). Moreover, in Brazil, around 70% of the municipalities are smaller than 50,000 hab. This means that incineration cannot be feasible. The idea of organizing groups of municipalities (that could be a partial solution) is not discussed by aiuthors.
Moreover, it is necessary to discuss the solution for such small/medium municipalities must be discussed. Authors mention gasification process but does not present update information about the gasification plants now under construction. There are other plants besides those mentioned in the paper.
Author Response
Dear reviewer,
All authors are grateful for their notes and comments. We consider all of them very important for the improvement of our manuscript.
We understand all placements and strive for full compliance or justification.
Next, we present the answers corresponding to the changes made to the manuscript.
Considering the first note made, "Authors discusses mainly incineration processes. In fact, this is a process already in place in several industrialized countries and the Brazilian environmental legislation addresses is in an adequate way. However, authors fail when discussing economic feasibility aspects. It does not seem incineration process is feasible for plants smaller than 600 t/d of MSW. This corresponds in average for municipalities with around 600,000 inhabitants (considering an average of 1.0 kg/hab/day). Moreover, in Brazil, around 70% of the municipalities are smaller than 50,000 hab. This means that incineration cannot be feasible. The idea of organizing groups of municipalities (that could be a partial solution) is not discussed by authors.”
We appreciate the alert and guide our actions toward adding paragraphs (lines 380 to 388) that seek to elucidate the issue by discussing the idea of consortium.
The second issue raised in the review was "Moreover, it is necessary to discuss the solution for such small/medium municipalities must be discussed. Authors mention gasification process but does not present update information about the gasification plants now under construction. There are other plants besides those mentioned in the paper”.
Again, we understand the point highlighted by the reviewer and, for assistance, a new paragraph was added (lines 320-323)
Thus, considering the changes made, we hope to have met your requests and we await your approval, or further comments, for the approval of the manuscript.
Yours sincerely,

Reviewer 3 Report
Mass burning was regarded as an available method to deal with MSW in this work, taking the global scenario into consideration, the possible application potential of such technologies in Brazil was preliminarily discussed by the authors, the work can be published after a few revision:
1. Reference numbers are not recommended for direct use as sentence elements, such as lines 123, 157, 188, 191, et al.
2. Some more recent references about the global scenario are required.
3. It is better to evaluate the technology maturity in the current research stage, especially in the Section 2.1, the detailed application of WtE should be clarified from the perspective of technologies, and then, the technology maturity can be further discussed. One more table is recommended to be added.
4. In the Section 2.2.3., LCA should be considered in the environmental issues, the references concerned should be added.
5. The sharpness of the picture needs to be improved (Especially figure 2,3,4).
Author Response
Dear reviewer,
All authors are grateful for their notes and comments. We consider all of them very important for the improvement of our manuscript.
We understand all placements and strive for full compliance or justification.
Next, we present the answers corresponding to the changes made to the manuscript.
In response to your first comment, “Reference numbers are not recommended for direct use as sentence elements, such as lines 123, 157, 188, 191, et al.”, we have to say that all citation forms have been corrected in the manuscript, on the indicated lines.
Then your second comment was “Some more recent references about the global scenario are required.” In response to your comment, we searched for European data, which have been updated, including Figure 2 which was inserted (lines 223-244). We inform you that no other more recent data were found in addition to those inserted in the text.
Continuing, your third comment was "It is better to evaluate the technology maturity in the current research stage, especially in the Section 2.1, the detailed application of WtE should be clarified from the perspective of technologies, and then, the technology maturity can be further discussed. One more table is recommended to be added." In response to it, the new subtopic 2.1 was created, exclusively, to meet your request (lines 77 - 183).
In response to your fourth comment "In the Section 2.2.3., LCA should be considered in the environmental issues, the references concerned should be added.” To respond to this request, a new paragraph was created by us (lines 598-602).
And finally, in response to your fifth and final note, “The sharpness of the picture needs to be improved (Especially figure 2,3,4)”, Our actions were acted on Figure 2, which was modified by a figure with more recent content, and figures 3 and 4, too, were modified for a clearer view.
Thus, considering the changes made, we hope to have met your requests and we await your approval, or further comments, for the approval of the manuscript.
Yours sincerely,

Round 2
Reviewer 1 Report
The authors addressed all the reviewer’s comments and the quality of the paper has increased. However, there are still some minor issues to be addressed:
1) Figure 2 has been improved but it still shows information only about European countries. It would be useful to add another figure about waste treatment in countries outside Europe. Then, the authors can refer to this figure while providing data in between lines 251-311.
2) The sentences of lines 241-244 are not clear and must be rephrased.
3) The resolution of Fig. 3 must be improved.
Author Response
Dear reviewer,
All authors are grateful for the new notes and comments. We consider all of them very important for the improvement of our manuscript.
We understand all placements and strive for full compliance or justification.
We also thank you for your comment on the evolution of the manuscript in the previous round.
Next, we present the answers corresponding to the new changes made to the manuscript.
By topic 1 pointed out:
Figure 2 has been replaced and improved, displaying information on more countries, besides Europe. Then we refer to this figure 2 in the passage where we provide data between lines 251-311.
The text was edited between lines 231 to 237, due to it referring to figure 2 previously used. As it has been replaced to display more countries, the text was also adjusted.
By topic 2 pointed out:
The sentences between lines 241-244 were rewritten to improve the understanding that the authors wanted to highlight.
By topic 3 pointed out:
The resolution of Figure 3 was improved as requested.
All new changes in the text are highlighted in blue for a better location, as the lines observed by the reviewer have been subjected.
Thus, considering the changes made, we hope to have met your requests and we await your approval, or further comments, for the approval of the manuscript.
Yours sincerely,
